# Potentially Toxic Elements (PTEs) Composition and Human Health Risk Assessment of PM10 on the Roadways of Industrial Complexes in South Korea

**Jin-Young Choi** [1,*,†], **Hyeryeong Jeong** [1,2,†], **Kongtae Ra** [1,2] **and Kyung-Tae Kim** [1]

[1] Marine Environmental Research Center, Korea Institute of Ocean Science and Technology (KIOST), Busan 49111, Korea; hrjeong@kiost.ac.kr (H.J.); ktra@kiost.ac.kr (K.R.); ktkim@kiost.ac.kr (K.-T.K.)

[2] Department of Ocean Science (Oceanography), KIOST School, University of Science and Technology (UST), Daejeon 34113, Korea

* Correspondence: jychoi@kiost.ac.kr; Tel.: +82-51-664-3216

† These authors contributed equally.

**Abstract:** Road and industrial origin particulate matters (PM) are a significant source of potentially toxic elements (PTEs), with health risks to the surrounding residents. In Korea for 60 years, although industries, roads and automobiles have increased aggressively, there are still few PTEs data in PM in road-deposited sediment (RDS) of industrial complexes (ICs). Therefore, this study aimed to investigate the PTE composition of on-road PM10 from nine major ICs and its pollution degree in Korea and evaluate its human health risks. The geo-accumulation index ($I_{geo}$) and pollution load index (PLI) elucidated that on-road PM10 were severely polluted by Sb, Zn, Ag and Pb. A combination of principal component analysis (PCA) and chemical tracers was used to define the PTEs sources. The results showed that non-exhaust emission from vehicles' activity is the primary source of PTEs in on-road PM10, and industrial emissions are the secondary source. The riskiest pathway on carcinogenic and non-carcinogenic by on-road PM10 with PTEs was in-gestion. Traffic origin PTEs including Pb, As, Sb and Cd had a more significant impact on carcinogenic and non-carcinogenic health than those of industrial origins. These results could help mitigate public health risks arising from on-road PM10 and improve air quality in ICs.

**Keywords:** road-deposited sediment (RDS); PM10; non-exhaust emission; antimony (Sb); non-cancer risk; cancer risk

## 1. Introduction

Rapid industrialization has increased the number of road and transport vehicles and has generated polluting road-deposited sediments [1,2]. Road-deposited sediments (RDS) are a complex mixture of surrounding soil-derived materials, airborne particles, pesticides, metals, organic materials and particles from traffic-related wear on tires, brakes, roads and paints [3–6]. Potentially toxic elements (PTEs) in RDS are of major concern because traffic-related road dust is enriched with nondegradable toxic metals [7–10].

The one of major sources of RDS are exhaust and non-exhaust emissions from road and vehicle wear [11,12]. In busy road areas, Traffic activities can contribute to 90% of ambient particulate matter (PM) [13]. There is a growing trend in non-exhaust emission rates [14,15], and various studies have elucidated that the major sources of PTEs such as Zn, Cu, Pb, Cd and Cr are the road non-exhaust emissions [16–20]. Because pollution levels of PM from traffic activities mostly exceed EU standards for human health [21,22], many countries have made considerable efforts to control on-road PM and develop technologies, such as encouraging the development of electric vehicles, to lose weight of the vehicle emissions [11,14,21,22]. However, Timmers and Achten [11] found that non-exhaust wear emissions account for 90% of PM10 and forecasted that the proportion is likely to increase

in future as vehicles become heavier with electric batteries. This means that even if the emission control policy is strengthened, RDSs will not be free from such pollution.

PM10 in RDS (On-road PM) is potentially fugitive dust, readily suspended with long residence time in ambient air, and it has adverse effects on residents and road users [23,24]. PM10 can travel hundreds of kilometers [25]. It is proved that PM from traffic activities is harmful to human beings and surroundings and strongly correlates with early mortality [26]. PTEs such as As, Cr, Sb and Pb attached to PM can also contaminate plant tissues [27,28] and interfere with their physiological processes [29,30].

Identifying the composition and origin of PM in RDS is a key step in environmental policy measures to control PM emissions by mitigating on-road PM. In the EU, federations are jointly working to reduce emissions and non-exhaust emissions from transportation activities through the Convention on Long Range Trans boundary Air Pollution (CLR-TAP) [14,15]. In South Korea, the length of roads constructed increased by approximately 400% to 11,714 km in 2018, and the number of cars increased 703 times from 1960 to 2016 [31]. According to a report on air pollutant emissions in Korea, the proportion of PM10 on the road originating from traffic reached 42.6% in ambient air [32]. Still, this value did not include non-exhaust emissions. In Korea, only exhaust emissions of vehicles are only regulated except non-exhaust emissions [33].

Atmospheric PM is also deposited on the road. Industrialization in Korea has rapidly occurred since the 1960s. Environmental pollution is closely related to this economic growth and increase in production [34]. The high growth rate in South Korea and Asia has increased anthropogenic pollutant emissions with insufficient control [35]. Korea has become the fifth largest steel producer [36], the fourth largest ethylene producer [37] and the tenth largest economy through rapid industrial growth focused on the steel, petrochemical, automotive, shipbuilding and machinery industries [38,39]. Metals, including PTEs emitted from industrial facilities, migrate several kilometers [40] and are supplied to residential areas and roads [41]. The strenuous activity of heavy vehicles (HDVs) in the IC area can promote PM generation on roads [42].

In Korea, the national industrial complex (IC) is an area of industrial facilities and related education, housing, culture and public green spaces [43]. Therefore, research on-road PM pollution in industrial areas is important for human health in Korea. Our previous research reported that roads in IC in Korea are heavily polluted with PTEs such as Cu, Zn, Cd and Pb [44], and the fine RDS (less than 63–125 μm) contain high levels of PTEs (i.e., Zn, Cu, Pb and Sb) [19,45,46].

Improved analytical technology and focused recent research have proven that the primary source of PTEs on the road is vehicle activities [12,47–49]. There are higher PTE concentrations in the finer particles of RDS [50], and traffic activities can serve air quality adverse effects with suspend-able RDS [51]. Therefore, to protect the actual impact on human health from industrialization and consequent increase in roads and vehicles, it is essential to study RDS for particles with an upper size limit (USL) of 10 μm [51,52]. Even though public health risk assessment using RDS has been actively carried out for a decade, there are still few reports from studies on PM of RDS as shown in Table 1. Our previous study showed that the PTE level of RDS in an industrial complex in Korea is at a level that has a severe impact on the surrounding ecological environment [44], and the proportion of fine particles is high. We found that public health assessments for PM are required.

Therefore, as a follow-up study, we performed PTE pollution level, source allocation and human health risk assessment for the fly fraction of PM10 in road dust in this area. The results of this study will provide information on the need for road dust control in industrial complexes and the target sources for this.

**Table 1.** Previous RDS studies in recent a decade which are dealt with the assessment of human health risk by RDS with pollutants, and the suitable research scope with its particle size fraction.

| Upper Size Limit (μm) | Land Type (City) | Reference (Year) | Suitable Research Scope with this Particle Size Fraction |
|---|---|---|---|
| 2000 | Urban (Tehran) | Kamani et al., 2017 [53] | Pollution level of study area and human health risk from RDS including ingestion and dermal contact |
| | Urban and Industrial areas (Queensland, Australia) | Ma et al., 2021 [54] | |
| 1000 | industrial park (Baoji) | Wang et al., 2014 [55] | |
| 850 | Urban (Maha Sarakham) | Ma and Singhirunnusorn, 2012 [56] | |
| | Urban (Shanghai) | Shi et al., 2010 [57] | |
| 100–150 | Urban (Zhengzhou) | Wang et al 2020 [58] | |
| | Urban (Zhengzhou) | Ferreira-Baptista and de Miguel, 2005 [59] | |
| 89 | Urban (7 cities in China) | Yang et al., 2020 [49] | Stormwater runoff with RDS and environmental and human health risks from RDS (including ingestion and dermal contact) |
| 75 | Urban (Gejiu) | Guo et al., 2021 [60] | |
| | Urban (Xi'an) | Wei et al., 2015 [61] | |
| 63 | Industrial area (Tianjin) | Hu et al., 2016 [62] | |
| | Urban (Tehran) | Saeedi et al, 2012 [63] | |
| | Urban (Nanjing) | Li et al., 2013 [64] | |
| | Urban (Petaling Jaya) | Shabanda et al., 2019 [65] | |
| | Urban (Chelyabinsk) | Krupnova et al., 2020 [66] | |
| 25 | Urban (Xinxiang) | Cao et al 2017 [23] | |
| 10 | Urban (Ahvaz) | Najmeddin and Keshavarzi 2018 [67] | The relationship between air quality and RDS, human health risks from RDS including ingestion, dermal contact and inhalation |

## 2. Materials and Methods

### 2.1. Study Areas

Korea has experienced rapid industrial growth since the 1960s. Currently, there are approximately 1,241 large and small ICs, of which 47 are national ICs [68]. In this case, 60% of factory sites in Korea belong to major national ICs, and this figure has increased over the past 10 years [69].

In this study, PTEs from PM10 in the RDS from nine ICs in Korea were investigated. The locations and information of the nine ICs are summarized in Figure 1 and Table 2, respectively. The investigated ICs were Sihwa (SH), Busam (BS, Noksan), Gunsan (GS), Changwon (CW), Daebul (DB, Yeoungam), Pohang (PH), Ulsan (US), Gwangyang (GY) and Onsan (OS); they were selected to ensure sampling of representative industries of South Korea. Various studies conducted from 1990 to 2020 revealed that major ICs in Korea, such as Pohang and Ulsan, polluted their surroundings and residential areas with organic pollutants or PTEs, causing public health and social problems [33,70–72]. From 2010 to 2014, the average PM10 level in the ambient air near the investigated ICs was 42–52 μg/m$^3$ per year, exceeding the annual PM10 permissible level set by the world health organization (20 μg/m$^3$) [73]. The number of employees in the ICs investigated in this study ranges from approximately 8,000 to 300,000, and there will likely be many road users in these areas [44]. Moreover, the ICs are based an average of 14.25 km away from residential areas with infrastructure, and 45% are located within 10 km from residential areas [69]. Thus, PM10 on roads in the IC may have an adverse effect on the health of workers and residents of the IC.

**Table 2.** Details and information of nine evaluated industrial complexes in Korea.

| Name of Industrial Complex (Abbreviation) | Major Industries | Established Year | Area for Industrial Facilities [1] | Employee [1] | Tota Production [1] | Estimated Unit Rate for Traffic with Industrial Trucks [2] | Location | | Numbers of Samples |
|---|---|---|---|---|---|---|---|---|---|
| | | | 1000 m$^2$ | Persons | 100 Million Won | Trucks/1000m$^2$ | Longitude | Latitude | N |
| Shihwa (SH) | machine and fabricated metal | 1977~ | 30,800 [43] | 292,070 [43] | 8,833,926 | 1.09 | 37°18′ N | 126°46′ E | 25 |
| Bussan: Myeongji·Noksan (BS) | steel manufacture and automotive | 1990~2007 | 4317 | 35,702 | 2,654,877 | 0.26 | 35°05′ N | 128°51′ E | 19 |
| Gunsan (GS) | shipbuilding | 1988~1994 | 5577 | 8,189 | 1,599,063 | - | 35°57′ N | 126°35′ E | 12 |
| Chagnwon (CW) | steel manufacture, machine | 1974~ | 17,242 | 93,094 | 12,292,492 | - | 35°12′ N | 128°39′ E | 15 |
| Daebul (DB) | Shipbuilding | 1989~1997 | 6640 | 13,738 | 768,158 | 0.12 | 34°46′ N | 126°27′ E | 14 |
| Pohang (PH) | iron manufacture | 1990~ | 13,613 | 12,654 | 3,986,076 | 0.4 | 35°58′ N | 129°22′ E | 19 |
| Ulsan (US) | Petrochemical, fertilizer, Shipbuilding and automotive | 1991~ | 33,771 | 101,415 | 36,157,916 | 0.26 | 35°32′ N | 129°20′ E | 26 |
| Gwangyang (GY) | iron manufacture | 1992~ | 21,833 | 11,354 | 4,738,990 | 0.12 | 34°56′ N | 127°45′ E | 21 |
| Onsan (OS) | nonferrous metal processing and smelting Industry | 1993~ | 16,573 | 16,306 | 12,937,634 | - | 35°26′ N | 129°20′ N | 14 |

[1] Korea Industrial complex corporation 2013. [2] Korea Transport Institute 2009: Estimated unit rate = Traffic volume of nearby branches of the industrial complex/unit area of the industrial complex.

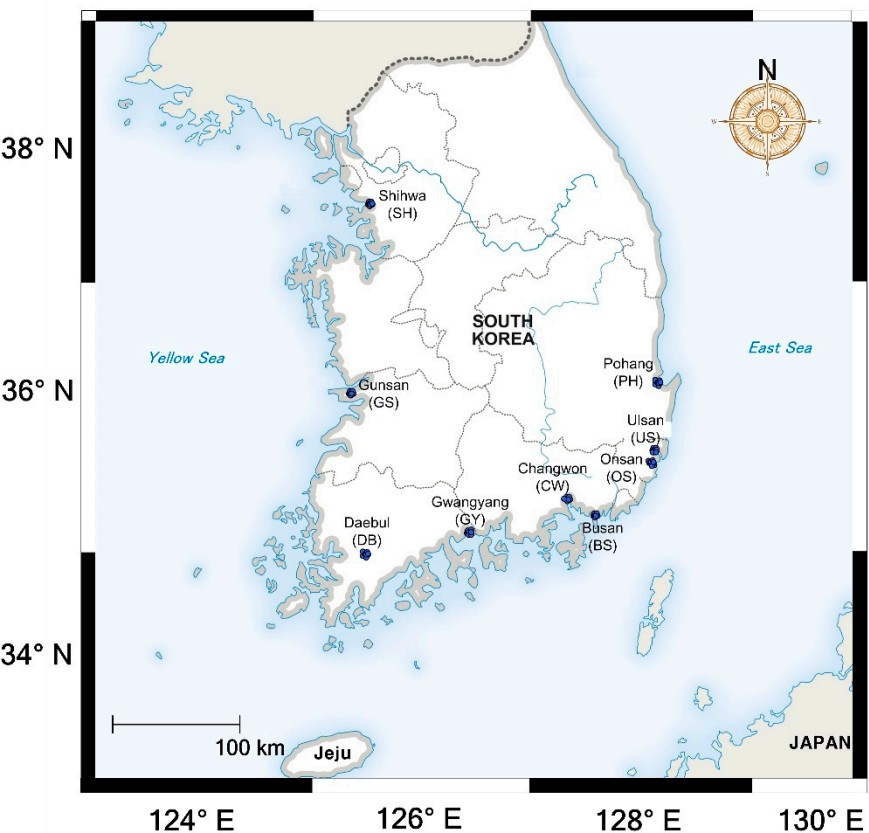

**Figure 1.** Map and sampling stations of nine industrial complexes where RDS was collected in this study. PM10 samples from in this study were separated from bulk RDS samples of our previous study [44]. More detailed information of the study area can be obtained from the previously reported study [44].

### 2.2. Sample Collection and Separation of PM10

This study was conducted on samples obtained from the bulk RDS samples collected in our previous study [44], for the follow-up study. More detailed information of the study area can be obtained from the previously reported study [44]. A total of 165 RDS samples were collected from nine ICs. In this case, 14 to 25 sampling points were selected on the main roads in each IC at equal intervals to reflect the RDS pollution from each IC. Sampling campaigns were performed in December 2013, after a continuous period of no rainfall for seven days. The dry vacuum cleaning method has been adopted in several studies on road dust to collect the finest particles from the road surface [7,20,74,75]. In this case, 95% of the total road RDS is accumulated on the road within 1 m of the margin edge the road, including the curb [76]. Therefore, RDS samples were collected from the road curve surface using a dry vacuum system (Dyson DC35, Dyson James Ltd., USA) with plastic brushes at each sampling point, and each sample was collected several times from 0.25 m$^2$ and composited to ensure representativeness. RDS was collected as much as possible until there were no particles remaining on the road surface at each sampling point, and the inner space of the vacuum cleaners and plastic brush was cleaned or changed after every single sampling to avoid cross-contamination between samples from other sampling points. The collected RDS samples were placed in plastic bags and transported to the laboratory.

After air drying at ambient temperature, fine particles smaller than 10 μm were separated from each RDS sample to collect PM10 using a nylon sieve and a vibratory sieve shaker (Analysette 3, Fritsch Co., Germany). To separate PM10 from the bulk RDS samples without loss, a standard sieve of 10 micrometers and was used a vibratory sieve shaker. More than five elastic silicone balls were placed in a standard sieve of 10 micrometers to operate automatic vibration using a vibratory sieve shaker. After waiting for the filtered

PM10 particles to settle down from resuspension, they were carefully transferred to a plastic vial using a plastic brush from the sieve. At all sampling points, the average mass of PM10 per 1 $m^2$ was $8.56 \pm 7.72$ g/$m^2$, and the relative concentration of PM10 fraction of the total RDS was $1.18 \pm 0.83\%$ (Table S1).

*2.3. Analytical Techniques and QA/QC*

About 0.1 g of PM10 sample was totally digested, dried on a hot plate, and re-dissolved in 1% $HNO_3$ at 180 °C for 36 h. Mixed acid (HF, $HNO_3$ and $HClO_4$) of a pure grade or higher was placed in an acid-cleaned Teflon vessel. The sediment samples were analyzed for PTEs, including Al, Fe, Li, Mg, P, Ti, Ag, As, Be, Cd, Co, Cr, Cu, Hg, Mn, Ni, Pb, Sb, Sn, V and Zn using inductively coupled plasma mass spectrometry (ICP-MS; Elemental-X7, Thermo Scientific, USA). Hg was analyzed using an automatic mercury analyzer (Hydra-C, Teledyne, Canada). To control data quality, the blanks and triplicate determinations were performed, and all lab-ware was acid cleaned using $HNO_3$, HCl and Mili-Q before being used. All analytical procedure was performed in the cleanroom. Data accuracy was checked sixth times of the same analytical procedure using two types of certified reference materials, MESS-4 and PACS-3 (National Research Council, Ottawa, ON, Canada). The results indicated a recovery rate of 92.2–116.6% (Table S2).

*2.4. Pollution Evaluation Methods*

2.4.1. Geo-Accumulation Index ($I_{geo}$)

The geo-accumulation index ($I_{geo}$) was used to assess the pollution levels of PTEs in PM10 of the RDS. $I_{geo}$ is a method proposed by Müller [77] for evaluating the degree of metal contamination in sediments. This index indicates the non-natural concentration of each metal element in the sediment. $I_{geo}$ values were calculated using the following equation:

$$I_{geo} = \log_2\left(\frac{C_n}{k \times B_n}\right) \tag{1}$$

where $C_n$ and $B_n$ are the concentrations of metal in the soil sample and the background, respectively. K was 1.5, which was used to analyze natural fluctuations in the content of a given substance in the environment and to detect small anthropogenic influences [8].

$I_{geo}$ is generally classified into six categories, which are described as follows: $I_{geo} < 0$ (background concentration); $0 < I_{geo} \le 1$ (unpolluted to moderately polluted); $1 < I_{geo} \le 2$ (moderately to unpolluted); $2 < I_{geo} \le 3$ (moderately to highly polluted); $3 < I_{geo} \le 4$ (highly polluted); $4 < I_{geo} \le 5$ (highly to very highly polluted); and $I_{geo} > 5$ (very highly polluted [77]). In this study, the background values used to calculate $I_{geo}$ were the average crustal concentration [78] and the background levels of Korean soil [79,80].

2.4.2. Pollution Load Index (PLI)

It is not possible to assess the overall contamination level of PTEs using the $I_{geo}$ present in the RDS. Therefore, to evaluate the overall contamination level of PTEs analyzed in RDS, we used the pollution load index (PLI), proposed by Tomlinson et al. [81] as an index to evaluate the overall toxic effect of PTE levels in sediments. PLI has also been used to assess the level of PTE contamination in many other studies. The PLI values were calculated using the following equations:

$$PLI = \sqrt[n]{CF_1 \times CF_2 \times CF_3 \cdots \times CF_n} \tag{2}$$

The CF value is the ratio of the concentration of each metal element to the background concentration, and the background values used were the average crustal concentration [78] and the background levels of Korean soil [79,80]. The PLI is generally classified into seven categories, which are described as follows: PLI = 0 (background concentration), $0 < PLI \le 1$ (unpolluted), $1 < PLI \le 2$ (moderately to unpolluted); $2 < PLI \le 3$ (moderately polluted);

3 < PLI ≤ 4 (moderately to highly polluted); 4 < PLI ≤ 5 (highly polluted); and PLI > 5 (very highly polluted).

*2.5. Health Risk Assessment*

2.5.1. Non-Carcinogenic Risk Assessment

The non-carcinogenic and carcinogenic risks of PTEs in PM10 of RDS were calculated according to a health risk assessment model derived and advanced by the USEPA [82,83]. Ingestion, dermal contact and inhalation are the three major pathways of RDS exposure in humans [84–86]. According to the exposure factor handbook [63], the average daily dose (ADD, mg/kg/day) of PTEs due to PM10 in RDS exposure through ingestion ($\text{ADD}_{\text{ing}}$), dermal contact ($\text{ADD}_{\text{derm}}$) and inhalation ($\text{ADD}_{\text{inh}}$) can be estimated using the following equations, respectively:

$$\text{ADD}_{\text{ing}} = \frac{(C_i \times \text{IngR} \times \text{EF} \times \text{ED} \times \text{CF})}{\text{BW} \times \text{AT}} \tag{3}$$

$$\text{ADD}_{\text{derm}} = \frac{(C_i \times \text{SA} \times \text{AF} \times \text{ABS} \times \text{EF} \times \text{ED} \times \text{CF})}{\text{BW} \times \text{AT}} \tag{4}$$

$$\text{ADD}_{\text{inh}} = \frac{(C_i \times \text{IngR} \times \text{EF} \times \text{ED})}{\text{PEF} \times \text{BW} \times \text{AT}} \tag{5}$$

where $C_i$ (mg/kg) is the concentration of PTEs in PM10 RDS, IngR is the ingestion rate of PM10 in RDS, InhR is the inhalation rate (20 m$^3$/day), EF is the exposure frequency (365 days/year), ED is the exposure duration (years), CF is the conversion factor ($10^{-6}$ kg/mg), SL is the skin adherence factor (0.2 cm$^2$/h), SA is the skin surface area that comes in contact with RDS (5700 cm$^2$). ABS is the dermal absorption factor ($10^{-3}$); BW is body weight (70 kg for adults); AT is the average lifetime (in days, ED × 365); and PEF is the particle emission factor (1.316 × 10$^9$ m$^3$). The parameters necessary for the assessment of ADD were reported by Adamiec and Jarosz-Krzemińska; USEPA; Zhou et al.; and Jose and Srimuruganandam [84,85,87,88].

The potential non-carcinogenic risk of specific PTEs was assessed using the hazard quotient (HQ) according to the following equation [84]. HQ represents the ratio of ADD to the specific reference dose (RfD).

$$\text{HQ}_{\text{ing}} = \frac{\text{ADD}_{\text{ing}}}{\text{RfD}_{\text{ing}}} \tag{6}$$

$$\text{HQ}_{\text{derm}} = \frac{\text{ADD}_{\text{derm}}}{\text{GIABS} \times \text{RfD}_{\text{derm}}} \tag{7}$$

$$\text{HQ}_{\text{inh}} = \frac{\text{ADD}_{\text{inh}}}{\text{RfD}_{\text{inh}} \times 1000 \, \mu\text{g/mg}} \tag{8}$$

where $\text{RfD}_{\text{ing}}$ is the oral reference dose (mg/kg per day), $\text{RfD}_{\text{derm}}$ is the inhalation reference concentration (mg/kg per day), $\text{RfD}_{\text{inh}}$ is the inhalation reference concentration (mg/m$^3$) and GIABS is the gastrointestinal absorption factor. If the HQ value is ≤1, the health risk is very low or negligible. If HQ is >1, potential adverse effects may occur. The RfD was adopted from the Integrated Risk Information System (IRIS) database published by the US EPA.

The hazard index (HI) is the total risk of specific PTEs through multiple exposures and can be estimated using the following equation:

$$\text{HI} = \sum \text{HQ}_i \tag{9}$$

where i represents a unique exposure pathway. HI <1 means that there is no significant risk, whereas HI >1 means that chronic risk is more likely.

2.5.2. Carcinogenic Risk Assessment

The carcinogenic risk (CR) for individual pathways is determined using the ADD multiplied by the respective slope factor, according to the following equations:

$$CR_{ing} = ADD_{ing} \times SF \tag{10}$$

$$CR_{derm} = ADD_{derm} \times \left[ \frac{SF_o}{GIABS} \right] \tag{11}$$

$$CR_{inh} = ADD_{inh} \times IUR \tag{12}$$

where SF is the slope factor, $SF_o$ is the oral slope factor (mg/kg per day), and IUR is the inhalation unit risk ($\mu g/m^3$). If the CR is less than $10^{-6}$, the risk is negligible. However, if the CR is higher than the threshold value of $10^{-4}$, the risk is considered unacceptable because of the high carcinogenic risk to humans, according to the U.S. Environmental Protection Agency (US EPA). The chemical-specific parameters used in this model are listed in Table S3.

*2.6. Statistical Analysis*

PASW Statistics 18 was used for statistical analyses of the data. Analysis of variance (ANOVA) and nonparametric tests were conducted to compare the sample groups, and one-sample t-tests were used to test the independence of each group. Pearson's correlation analysis was used to correlate the data with confirmed normality, and Kendall's tau-b method was used to correlate the data without normality. Principal component analysis (PCA) was used to identify PM sources [58,89–92]. For factor analysis, factors with a minimum eigenvalue greater than 1 (initial eigenvalue > 1) were selected, and the Varimax rotation component value was used to clarify the relationships among various factors.

**3. Results and Discussion**
*3.1. PTE Contamination in PM10 of Road-Deposited Sediment (RDS)*
3.1.1. Concentrations of PTEs

The concentrations of major and trace PTEs in the PM10 of the RDS (on-road PM10) from the nine ICs are summarized in Table 3. The mean concentrations of major elements (Li, Fe, Al, Mg and Ti) were approximately 1–2 times the background levels of each element, and P was three times the background level. The abundance of major elements decreased in the order Li > Fe > Al >> Mg > Ti > P. The variability in concentration of the major elements of PM10 in the RDS as coefficient of variation (CV) was greatest for Fe (53%) and Ti (45%). At the maximum level, Fe was seven times, and Ti two times greater than the background concentration. In general, major elements such as Fe, Al and Ti in dust are soil-originated components [85,93,94]. However, this variability of Fe and Ti indicates that there is some anthropogenic input. Fe may be influenced by the steel industry and traffic activity with brake lining. Fe is the most abundant element in brake linings [95], and the content may vary depending on the type of lining [12,96,97]. Titanium dioxide nanomaterials ($TiO_2$ NMs), the second most-produced nanomaterial worldwide [98], has been focused because it is released from these consumer products during their use phase and end-of-life and local environmental accumulation near nanomaterial manufacturing sites [99]. However, the high variability and concentrations compared to background of major elements in on-road PM10 at specific sites may represent significant on-road emissions of Fe and Ti from anthropogenic activities in those areas.

**Table 3.** Statistical characteristics of major and potentially toxic elements in PM10 of RDS from nine national industrial complexes in South Korea.

| Elements | | Unit | Min | Max | Median | Mean | SD | CV (%) | Background Level |
|---|---|---|---|---|---|---|---|---|---|
| Major elements | Al | % | 3.9 | 9.5 | 6.5 | 6.4 | 1.1 | 18 | 8.15 [79] |
| | Fe | | 2.9 | 27 | 7.3 | 8.4 | 4.5 | 53 | 3.88 [79] |
| | Li | | 12 | 97 | 30 | 32 | 11 | 34 | 24 [78] |
| | Mg | | 0.49 | 4.3 | 1.1 | 1.2 | 0.37 | 31 | 1.488 [79] |
| | P | | 0.041 | 0.35 | 0.094 | 0.11 | 0.047 | 45 | 0.033 [79] |
| | Ti | | 0.19 | 0.88 | 0.43 | 0.43 | 0.11 | 25 | 0.384 [79] |
| Trace elements (heavy metals, HMs) | Ag | mg/kg | 0.570 | 509 | 3.74 | 22.3 | 70.8 | 317 | 0.053 [78] |
| | As | | 7.0 | 6671 | 24 | 193 | 668 | 345 | 6.83 [80] |
| | Be | | 0.96 | 4.0 | 2.2 | 2.3 | 0.61 | 27 | 2.1 [78] |
| | Cd | | 0.889 | 927 | 6.37 | 29.6 | 107 | 361 | 0.287 [79] |
| | Co | | 7.12 | 713 | 24.0 | 33.8 | 60.7 | 180 | 12.99 [79] |
| | Cr | | 57.2 | 19224 | 351 | 894 | 1843 | 206 | 25.36 [79] |
| | Cu | | 54.5 | 19776 | 446 | 1223 | 2703 | 221 | 15.26 [79] |
| | Hg | | 0.030 | 64 | 0.41 | 2.4 | 6.9 | 292 | 0.05 [78] |
| | Mn | | 744 | 18893 | 2020 | 2915 | 2545 | 87 | 774 [78] |
| | Ni | | 20.5 | 9192 | 144 | 258 | 727 | 282 | 17.68 [79] |
| | Pb | | 62.61 | 47827 | 695 | 2539 | 6380 | 251 | 18.43 [79] |
| | Sb | | 3.32 | 3476 | 39.5 | 161 | 460 | 286 | 0.46 [78] |
| | Sn | | 6.09 | 639 | 45.1 | 71.4 | 87.4 | 122 | 2.1 [79] |
| | V | | 48.3 | 656 | 95.9 | 103 | 52.4 | 51 | 44 [79] |
| | Zn | | 629.26 | 169955 | 3868.6 | 7501.2 | 15716 | 210 | 54.27 [79] |

The mean concentrations of 17 PTEs determined in on-road PM10 decreased in the following order: Zn > Mn > Pb > Cu > Cr > Ni > V > Sn > Sb > As > Co > Cd > Ag > Be > Hg. The average concentration ranged widely from 1–3 times the background (Be, Co, Sr, Tl, V) to 350–420 times the background (Sb, Ag). This observed PTEs order is also commonly reported in other previous research [52,54,58]. Zn, Mn, Pb, Cu, Cr and Ni are mainly the significant contributors to the wear of the natural soil and asphalt, and Zn, Pb, Cu, Sb and Cd are found most predominantly in road [54]. The most abundant PTE in on-road PM10 was Zn, with a concentration range of 629–169955 mg/kg. The mean concentration of Zn was 7501 mg/kg, which accounted for 48% of the mass of all trace metals. The main source of Zn in RDS or road dust is tires [54,100,101]. Zinc is added to tires for acceleration of sulfur vulcanization and resistance to degradation by ultraviolet radiation and accounts for approximately 1% of the weight of the tire tread [102]. A previous study reported that $8.7 \pm 2.0$ g/kg of zinc was detected in various tire powder samples, with the most homogeneous concentration (RSD 23%) compared to other metals [47]. For this reason, Zn is recommended as the most suitable metal element marker for tire and road wear particles (TRWP) [47]. Many previous studies have shown that Zn is the most abundant metal element in RDS in Korea without considering particle size [20,44,75]. The elements with the next highest concentrations were Mn, Pb and Cu, with averages of 2915, 2539 and 1223 mg/kg, respectively. Cr, Sr, Ni, As, Sb and V were also relatively high in concentration, ranging from 103 to 894 mg/kg in all PM10 samples. Notably, the Sb value reached 350 times that of the background level in Korean soil [80]. Antimony levels in the environment have been increasing due to industrial use, biomass burning and vehicle origin, such as brake pads [103–105]. Metal concentrations in on-road PM10 investigated in this study were compared with previously reported data in other studies (Table 4). The concentrations of PTEs in on-road PM10 on roads from ICs were generally higher than those on roads from urban areas that were reported to be contaminated. Emissions from the steel and smelting industries are the leading causes of anthropogenic PTE supply in the environment [106,107], and active HDV in ICs may also be another cause of high PTE levels [42].

**Table 4.** Mean levels of HMs concentrations in PM10 of RDS in several previous studies. (unit. mg/kg).

| City, Country | Characterization | Size μm | Zn | Cu | Pb | Sb | Cr | Hg | Ni | Cd | As | V | Reference |
|---|---|---|---|---|---|---|---|---|---|---|---|---|---|
| 9 industrial cities, Korea | Industrial area, RDS | <10 | 7501.2 | 1223 | 2539 | 161 | 894 | 2.4 | 258 | 29.6 | 193 | 103 | This study |
| Fushun, China | Coal-based city, RDS | <10 | 174 | 115 | 50.7 | - | 5558 | 0 | 9 | 1.3 | 14 | 14 | Kong et al., 2012 [9] |
| Madrid, Sapin | Urban, RDS | <10 | 1135 | 444 | 121 | 82 | 56 | - | 139 | - | - | 13 | Karanasiou et al., 2014 [108] |
| Zürich, Switzerland | Urban, RDS | <10 | 2183 | 3547 | 247 | 324 | 330 | - | 504 | 10 | 19 | 56 | Amato et al. 2011 [15] |
| Barcelona, Spain | Urban, RDS | <10 | 1572 | 1332 | 248 | 196 | 229 | - | 58 | 3 | 12 | 84 | Amato et al. 2011 [15] |
| Girona, Spain | Urban, RDS | <10 | 1760 | 1055 | 128 | 64 | 188 | - | 191 | 2 | 11 | 54 | Amato et al. 2011 [15] |
| London, UK | Urban, ambient dust | <10 | 44.6 | 53.2 | - | 6.73 | - | - | - | - | - | 2.1 | Gietl et al., 2010 [109] |

3.1.2. Assessment of Pollution Degree with $I_{geo}$ and PLI

To assess the anthropogenic pollution levels of PTEs in the PM10 fraction of RDS, the pollution levels of individual PTEs were calculated and the overall pollution level of various PTEs with $I_{geo}$ and PLI in the investigated industrial cities were also calculated. The results are summarized in Table 5.

The mean $I_{geo}$ values of the major and PTEs in all analyzed PM10 samples decreased in the following order: Sb > Zn > Ag > Pb > Cu > Cd > Sn > Cr > Hg > Ni > As > Mn > P > V > Co > Fe > Li > Ti > Be > Al > Mg. Among these PTEs, Sb, Zn, Ag, Pb, Cu and Cd had

the highest $I_{geo}$ values. The mean $I_{geo}$ values of Sb, Zn, Ag and Pb were 6.2, 5.9, 5.8 and 5.0, respectively, indicating that these ICs were extremely contaminated. The mean $I_{geo}$ values of Cu and Cd were 4.5 and 4.1, respectively, indicating heavy to extreme contamination. The $I_{geo}$ values of Sn and Cr indicated heavy contamination, and Hg, Ni and As values indicated moderate to heavy contamination. The ICs we investigated were unpolluted to moderately contaminated with the remaining PTEs or had background concentrations of them. Sb, the most highly polluting PTE in the ICs we investigated, is recognized as a reliable tracer of brake wear, along with Cu [54,110]. Sb is a toxic metalloid that is emerging as a global environmental problem because of its widespread use and mining [48]. In the case of roads, Sb is emitted through disk wear of brake systems, and exceedingly high concentrations of Sb in fine particles (<125 μm) and runoff from urban roads in Korea have been reported [19,46]. The concentration of Sb in the brake pad generally used in European cars are in the range of 1–5% [111]. As mentioned in Section 3.1, Zn is a tire tread tracer, and a high $I_{geo}$ level of Zn might be caused by tire wear in industrial areas. Cu is a traditional brake pad tracer [112]. In California, brake pad Cu account for more than 60% of all Cu in urban watershed runoff [113]. Therefore, a reduction policy stating that Cu will be replaced by other elements in all brake pads and discs produced by 2025 has been implemented [100,112]. Lead pollution no longer occurs from fuel owing to the restrictions on fuel additions since 1993 in Korea. However, lead is still reported as a major pollutant in road dust in Korea and other countries [20,114]. Historical pollution can still be a problem, and researchers are now positing that it can occur from scales and construction paints, valance weight and batteries for vehicles [100,115]. Currently, high anthropogenic Ag pollution is frequently detected in the atmosphere, water and soil because of the wide industrial use of silver nanoparticles (NPs) [116–118]; however, there are still few reports on road sediment levels and sources. Therefore, further research is needed to assess the sources of on-road Pb and Ag. These results show that PM10 pollution on the roads of ICs in Korea was severely contaminated by tires and brake pads of vehicles and PTE-associated industries. Since it has been reported that PTEs can have adverse effects on the environment, and PM10 pollution is directly related to human health [28,96,119], it is important to determine the source of PM10 with PTEs to find an appropriate environmental solution.

**Table 5.** Summary of the pollution levels of individual PTEs and the overall pollution level of various PTEs with $I_{geo}$ and PLI in nine industrial cities.

| ICs | | SH | BS | GS | CW | DB | PH | US | GY | OS | Mean | Min | Max | SD | CV% |
|---|---|---|---|---|---|---|---|---|---|---|---|---|---|---|---|
| | Sb | 7.2 | 5.9 | 5.0 | 5.9 | 5.8 | 6.5 | 4.8 | 4.8 | 10.4 | 6.2 | 4.8 | 10.4 | 1.7 | 0.3 |
| | Zn | 5.4 | 6.7 | 4.8 | 5.7 | 6.0 | 5.1 | 5.5 | 5.5 | 8.1 | 5.9 | 4.8 | 8.1 | 1.0 | 0.2 |
| | Ag | 5.3 | 6.6 | 3.8 | 5.5 | 5.7 | 6.6 | 4.0 | 4.0 | 10.8 | 5.8 | 3.8 | 10.8 | 2.2 | 0.4 |
| | Pb | 6.2 | 5.5 | 4.0 | 4.5 | 4.4 | 5.0 | 3.2 | 3.2 | 8.8 | 5.0 | 3.2 | 8.8 | 1.7 | 0.4 |
| | Cu | 5.1 | 4.9 | 3.1 | 4.4 | 4.0 | 5.0 | 3.0 | 3.0 | 7.9 | 4.5 | 3.0 | 7.9 | 1.5 | 0.3 |
| | Cd | 3.6 | 4.9 | 2.3 | 3.8 | 4.4 | 4.3 | 2.7 | 2.7 | 8.7 | 4.1 | 2.3 | 8.7 | 1.9 | 0.5 |
| | Sn | 5.1 | 4.4 | 3.1 | 4.0 | 3.6 | 3.3 | 2.7 | 2.7 | 6.0 | 3.9 | 2.7 | 6.0 | 1.1 | 0.3 |
| | Cr | 3.3 | 5.4 | 2.9 | 4.3 | 5.8 | 2.2 | 3.4 | 3.4 | 2.7 | 3.7 | 2.2 | 5.8 | 1.2 | 0.3 |
| | Hg | 2.2 | 2.8 | 0.2 | 2.4 | 4.6 | 3.1 | 1.7 | 1.7 | 7.3 | 2.9 | 0.2 | 7.3 | 2.1 | 0.7 |
| | Ni | 2.6 | 4.2 | 1.5 | 2.8 | 3.0 | 1.4 | 3.0 | 3.0 | 2.7 | 2.7 | 1.4 | 4.2 | 0.8 | 0.3 |
| $I_{geo}$ | As | 1.5 | 1.4 | 0.9 | 0.7 | 2.0 | 2.9 | 0.8 | 0.8 | 7.0 | 2.0 | 0.7 | 7.0 | 2.0 | 1.0 |
| | Mn | 0.1 | 1.3 | 0.6 | 1.2 | 2.7 | 0.2 | 1.7 | 1.7 | 0.7 | 1.1 | 0.1 | 2.7 | 0.8 | 0.7 |
| | P | 1.5 | 1.4 | 1.3 | 0.9 | 0.8 | 1.1 | 0.5 | 0.5 | 0.5 | 1.0 | 0.5 | 1.5 | 0.4 | 0.4 |
| | V | 0.3 | 0.6 | 0.2 | 0.8 | 1.1 | 0.5 | 0.8 | 0.8 | 0.9 | 0.6 | 0.2 | 1.1 | 0.3 | 0.4 |
| | Co | 0.9 | 0.7 | 0.9 | 0.4 | 0.2 | 0.0 | 0.2 | 0.2 | 0.9 | 0.5 | 0.0 | 0.9 | 0.4 | 0.7 |
| | Fe | 0.3 | 0.8 | 0.0 | 0.6 | 1.2 | −0.3 | 0.7 | 0.7 | 0.1 | 0.5 | −0.3 | 1.2 | 0.5 | 1.0 |
| | Li | −0.1 | −0.4 | 0.3 | −0.1 | −0.5 | 0.0 | −0.5 | −0.5 | −0.8 | −0.3 | −0.8 | 0.3 | 0.3 | −1.1 |
| | Ti | −0.2 | −0.3 | −0.1 | −0.4 | −0.9 | −0.6 | −0.5 | −0.5 | −0.7 | −0.5 | −0.9 | −0.1 | 0.3 | −0.6 |
| | Be | 0.0 | −0.9 | −0.1 | −0.9 | −0.9 | −0.6 | −0.6 | −0.6 | −0.6 | −0.6 | −0.9 | 0.0 | 0.3 | −0.6 |
| | Al | −0.8 | −1.0 | −0.8 | −1.0 | −1.3 | −0.9 | −1.1 | −1.1 | −1.0 | −1.0 | −1.3 | −0.8 | 0.2 | −0.2 |
| | Mg | −1.1 | −1.0 | −1.0 | −0.8 | −0.6 | −1.1 | −0.6 | −0.6 | −1.3 | −0.9 | −1.3 | −0.6 | 0.3 | −0.3 |
| PLI | | 6.4 | 7.3 | 3.8 | 5.5 | 2.7 | 7.2 | 5.7 | 4.1 | 17.8 | 6.7 | 2.7 | 17.8 | 4.4 | 0.7 |

### 3.2. Source Identification of PTEs in PM10 from RDS

3.2.1. Source Apportionment of PTEs

PCA analysis was used to identify the source of on-road PM10 in the nine ICs (Table 6a,b). To classify the potential sources of PTEs in the PM10 of RDS, PCA was performed using varimax rotation with $I_{geo}$ data standardized with background concentration. The PCA results showed that there were four eigenvalues that were >1, and these four principal components explained 72.9% of the total variance (Table 6a).

**Table 6.** Results extracted by principal component analysis (PCA) for the geo-accumulation index ($I_{geo}$) of PTEs in the PM10 of RDS. (**a**) Initial eigenvalues, (**b**) rotated component matrix (PCA loadings >0.5 are shown in bold).

| (a) | | | |
|---|---|---|---|
| **Component** | **Initial Eigenvalues** | | |
| | **Total** | **% of Variance** | **Cumulative %** |
| 1 | 8.685 | 41.356 | 41.356 |
| 2 | 4.576 | 21.790 | 63.147 |
| 3 | 2.097 | 9.985 | 73.132 |
| 4 | 1.276 | 6.076 | 79.207 |

| (b) | | | |
|---|---|---|---|
| | **Component** | | |
| | **1** | **2** | **3** | **4** |
| Mg | −0.277 | **0.535** | −0.048 | **0.458** |
| Al | −0.145 | −0.407 | **0.803** | 0.074 |
| Ti | −0.148 | 0.025 | **0.775** | −0.188 |
| Fe | −0.012 | **0.843** | −0.362 | −0.027 |
| P | 0.083 | 0.166 | 0.137 | −0.756 |
| Li | −0.337 | −0.188 | **0.662** | −0.105 |
| Be | −0.132 | −0.372 | **0.738** | −0.100 |
| V | 0.278 | **0.464** | −0.131 | **0.681** |
| Cr | 0.055 | **0.874** | −0.286 | −0.036 |
| Mn | −0.172 | **0.676** | −0.491 | **0.365** |
| Co | 0.475 | **0.520** | 0.330 | −0.117 |
| Ni | 0.295 | **0.841** | −0.100 | −0.025 |
| Cu | **0.897** | −0.030 | −0.154 | −0.191 |
| Zn | **0.684** | 0.316 | −0.327 | −0.077 |
| As | **0.881** | −0.20 | −0.165 | 0.250 |
| Ag | **0.924** | 0.010 | −0.222 | 0.067 |
| Cd | **0.923** | 0.059 | −0.287 | 0.099 |
| Sn | **0.853** | 0.272 | 0.132 | −0.220 |
| Sb | **0.956** | −0.012 | −0.008 | −0.004 |
| Pb | **0.945** | 0.109 | 0.003 | −0.151 |
| Hg | **0.803** | 0.162 | −0.376 | 0.233 |

Principal component 1 from this result was mainly comprised of Cu, Zn, As, Ag, Cd, Sn, Sb, Pb and Hg, explaining 41.4% of the total variance (Table 6b). Among these PTEs, Zn, Pb and Cu were the most abundant PTEs in the PM10 of RDS collected from ICs, as shown in Section 3.1. Many previous studies have shown that Cu, Zn, Sb and Pb are elements released through representative non-exhaust emissions, such as vehicle parts and road wear [120]. Zn is related to tire wear, and Cu, Sb and Cd are mainly related to brake pad wear [54]. Brake pads were identified as the source of Sn in road dust, as reported by Mummullage et al. [121]. Due to these tracers, the PCA result implied that Cu, Zn, As, Ag, Cd, Sn, Sb, Pb and Hg derived from the same non-exhaust sources on roadways in the ICs in South Korea.

Principal component 2, which accounted for 21.8% of the total variance, was heavily comprised of Fe, Cr and Ni and moderately comprised Co, V, Mn and Mg (Table 6b). Even

though Cr, Ni, Co and V are related to asphalt, asphalt was excluded as a possible source because there was no correlation with Al, Li and Zn, which are important components of asphalt [121]. Principal component 2 was considered to represent an iron/steel manufacturing source because of the loading of Fe, Cr, Ni, Mn, V and Mg. A previous study reported that Cr and Ni were emitted from blast furnaces [6]. Mn is the most abundant metal in the steel industry sites in Korea because it is an important material for fabricating high-strength high-Mn steels. [122]. Vanadium is also added to steel to increase its elasticity, hardness and strength [123]. Mg, such as MgO, is mainly used as a refractory material in steel mills [124] and is a by-product of the casting process for high-Mn steel [33]. The significant positive correlation ($p < 0.01$) of these PTEs with Fe in PC2 supported that Fe, Cr, Ni, Co, V, Mn and Mg might originate from iron/steel manufacturing industries (Table S4).

We conclude that Al, Ti, Li and Be, which were found in principal component 3, originated from a natural source. They had a significant positive correlation ($p < 0.01$), indicating that they had the same source (Table S4). However, these PTEs were negatively correlated with the other elements, indicating that the other PTEs had different sources. Principal component 4 explained 6.076% of the total variance. PC4 was mainly dominated by Mg, V and Mn, which are related to the high-grade steel industry [33,122,123].

PCA was used again to identify pollution sources that affected the PM10 of the RDS and to classify the pollution characteristics by industry. The data used for PCA were the $I_{geo}$ values of PTEs identified as being of anthropogenic origin and pollution degree by each industrial complex using PLI. The results are shown in Figure 2a,b. For dimensional reduction, average values of $I_{geo}$ were used in the analysis. Principal component 1 was heavily loaded with PLI and PTEs (Cu, Zn, As, Ag, Cd, Sn, Sb, Pb and Hg) identified as products of traffic activity in Table 5b (Figure 2a). PC1 explained 60.357% of the total variance (Table S5), thus implying that PTEs from non-exhaust emissions of traffic activity may have strongly affected the pollution level of PM10 in RDS collected from the nine ICs. We classified the metal composition of on-road PM10 by industrial type using the scatter plot of PC1 and PC2 scores obtained from this analysis (Figure 2b). This revealed that ICs were classified into group 1 for PH, BS, CW and GY, and group 2 contained SH, US, GS, DB and OS owing to the metal composition and pollution characteristics of RDS PM10. The larger the factor score, the greater the pollution level. Group 1 (G1) contained major iron/steel ICs in Korea, which is the world's fifth largest steel producer [36]. Korea's largest steel companies (including POSCO and HYUNDAI) are in Pohang and Gwangyang [33], and there are large steelmakers in the BS(NS) in CW. Groups 2 (G2) and 3 (G3)are rich in petrochemical industries with fabricated metal industries for automotive and shipbuilding. In particular, Group 3, which only includes OS, is a large non-ferrous industrial complex with smelters that produce non-ferrous elements such as Ag and Ni [125] and includes a highly polluted area where the "Onsan Public Hazardous Disease" historically occurred owing to unknown industrial pollutants in the 1980s [126].

Figure 3 shows the PTE profiles for each group. The relative concentrations of Zn, Pb, Cu and Sb, which account for more than 50% of the total PTE inventory in PM10 of RDS, gradually increased toward G1, G2 and OS. In contrast, the relative concentrations of Mn, Cr and Ni gradually decreased toward G1, G2 and G3. When comparing G1 and G2, G2, a large non-ferrous industrial complex, had a large number of unit vehicles (mean value, G1 = 0.26, G2 = 0.49 cars/1000 m$^2$) (Table 2) and a high concentration of major non-exhaust metal elements (Zn, Cu, Pb, Sb). In summary, the main source of PM10 pollutants in RDS were vehicle activity and non-exhaust emissions. However, this result cannot divide the contribution affected the metal composition of on-road PM10 of facility origin atmospheric deposition and traffic origin non-exhaust particles.

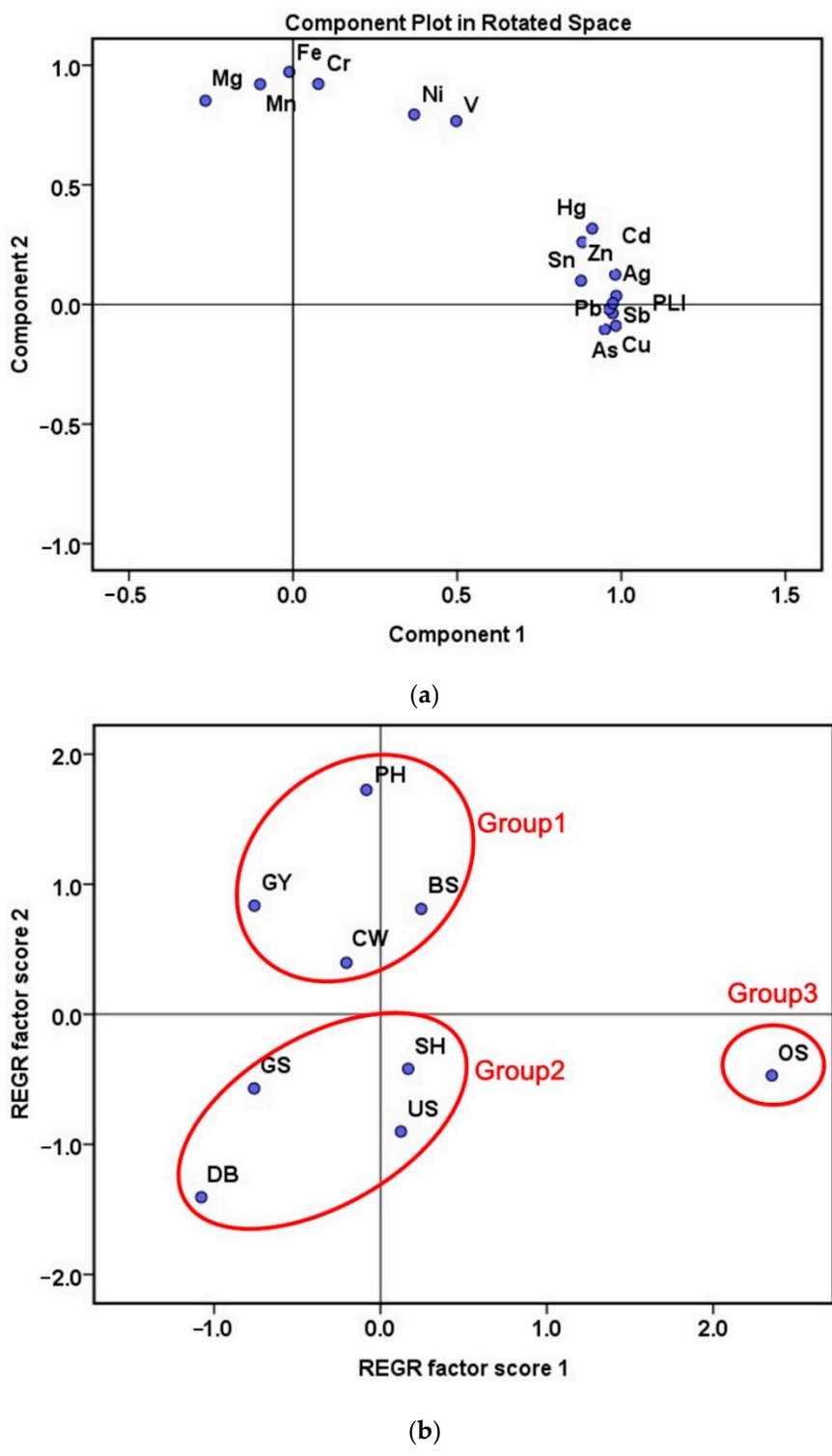

**Figure 2.** PCA results for identifying the pollution sources that mainly affect the PM10 of RDS and classification of the pollution characteristics by ICs. (**a**) Loading plot of the two principal components, (**b**) graph plotted with regression factors of PC1 and PC2.

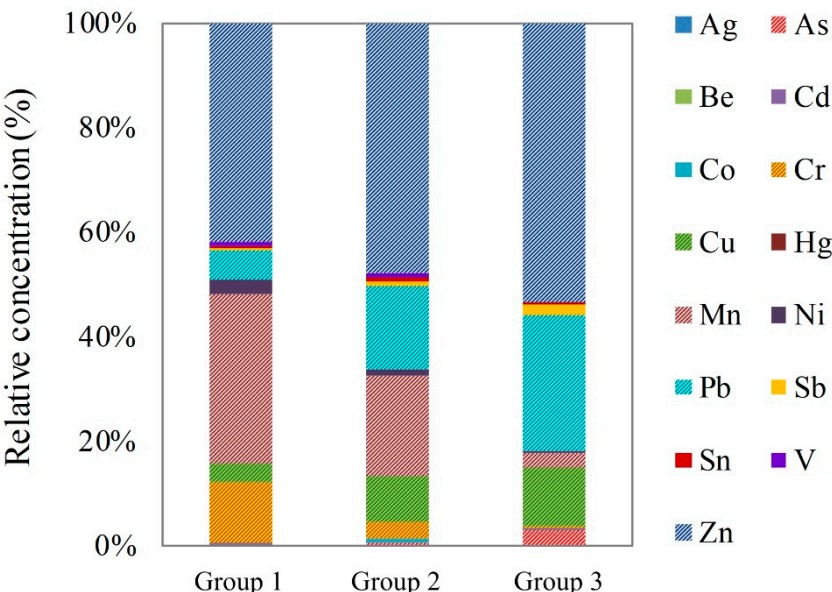

**Figure 3.** Heavy metal profile of PM10 in RDS from Group 1 (G1, steel industry sites), Group 2 (G2, fabricated metal industrial sites for automobiles and ship building, and petrochemical industrial sites), and Group 3 (G3, Onsan) (Non-ferrous smelting and petrochemical industrial sites).

### 3.2.2. Chemical Tracers of Non-Exhaust Emissions

The ratios of Zn, Cu and Sb in RDS are considered useful chemical tracers for brake wear-related emissions. Zn/Cu can be used to compare the contributions of the tire and brake systems.

Cu/Sb is a useful chemical tracer to confirm brake emissions. The values of these chemical tracers reported in the literature and from our investigation are listed in Table 7. Brake pad compositions normally show Cu/Sb values in the range of 9–18 [97]. The closer the Cu/Sb value of the RDS is to this value, the higher the possibility that the pollution source of road particles is the brake system. Our results show that the average Cu/Sb in PM10 of RDS is 7.1–22. This indicates that the Sb and Cu in PM10 in the RDS from the ICs depend on the load of the brake system emission. The results of this study are similar to those reported for roads in Spain [15]. The Zn/Cu ratio in road dust can inform the relative contribution of the tire and brake system, and the higher the ratio of Zn, the higher the ratio of tire wear particles in RDS [100]. Our results showed that the Zn/Cu ratio of on-road PM10 was close to the reported Zn/Cu values for tire dust or tire treads [16], suggesting a high contribution of tire wear to PM10 in industrial complexes.

Correlation analysis was performed for PLI, Zn/Cu and Cu/Sb to determine the relationship between the pollution level of PM10 in RDS and vehicle pollution (Table S6). The degree of PM10 pollution calculated by PLI showed a strong ($p < 0.001$) relationship with Zn/Cu and Cu/Sb, with correlation coefficients of −0.243 and −0.189, respectively, indicating that a higher contribution of the brake system and higher antimony content in the tire leads to a higher degree of anthropogenic PM10 pollution on roads.

In Delhi, non-exhaust emissions accounted for 86% of the area's road PM emissions, but exhaust emissions accounted for only 14% of on-road PM emissions [114]. According to the EU report, the annual emissions of PM10 from brake wear ranged from 0.29 (Norway) to 6.83 Gg (Germany) [127].

**Table 7.** Mean values of non-exhaust vehicle emission tracers from previous studies and this study. (-: not measured).

| City | Type of Area | Sample | Zn/Cu | Cu/Sb | Reference |
|---|---|---|---|---|---|
| Barcelona | Urban | RDS | 1.2 | 6.8 | |
| Zürich | Urban | RDS | 0.6 | 13.5 | Amato et al., 2011 [30] |
| Girona | Urban | RDS | 1.7 | 17.0 | |
| Fushun | Urban | RDS | 2 | - | Kong et al., 2012 [9] |
| London | School area | RDS | 3.5 | 7.1 | Gietl et al., 2010 [107] |
| | Urban | RDS | 0.8 | 7.9 | |
| SH | Industrial area | RDS | 5 | 10 | |
| BS | Industrial area | RDS | 13 | 21 | |
| GS | Industrial area | RDS | 16 | 19 | |
| CW | Industrial area | RDS | 9 | 12 | |
| DB | Industrial area | RDS | 17 | 23 | This study |
| PH | Industrial area | RDS | 16 | 11 | |
| US | Industrial area | RDS | 6 | 16 | |
| GY | Industrial area | RDS | 22 | 11 | |
| OS | Industrial area | RDS | 11 | 12 | |
| | Average | | 11 | 13 | |
| | Brake pad | | - | 9–18 | Iijima et al., 2007 [97] |
| | | | 0.13 | - | Westerlund, 1998 [95] |
| | Tier tread | | 14.5 | - | Adachi and Tainosho, 2004 [15] |
| | Tier dust | | 16.0 | - | |

In this case, 50% of the non-exhaust emission particles can be resuspended in ambient air, and they may be a secondary source of pollution that can accumulate on the road surface or near sidewalks [84]. Amato et al. [15] confirmed that the composition ratios of Cu and Sb in PM on roads and in ambient air were almost the same, suggesting that on-road PM10 could affect the surrounding atmosphere. Therefore, in this study, it was necessary to evaluate the health effects of PM10 in RDS on the workers and residents of ICs.

*3.3. Health Risk Assessment of PTEs in PM10 from RDS*

3.3.1. Non-Carcinogenic and Carcinogenic Risk

We assessed the non-carcinogenic and carcinogenic risk to humans of 9 PTEs through three pathways: ingestion, inhalation and dermal contact (Table 8, Figure 4). The results indicated that ingestion was the main pathway of PTEs into the local population in, followed by dermal contact and inhalation. This result is consistent with previously reported results for road dust and contaminated soil in other cities [84,85,128–130].

The estimated potential non-carcinogenic human health risks related to ingestion of PTEs from on-road PM10 are presented in ascending order, as shown in Figure 4a (Ti < Al < Fe < Sn>Hg < Be < Ag < Cd < Co < Li < V < Mn < Ni < Zn < Cu < Sb < As < Pb). In the case of ingestion exposure, the non-carcinogenic risks of Pb, Sb and As were the highest, and the risk levels of other PTEs were negligible.

Except for OS, the $HQ_{ing}$ calculated for PM10 of RDS in all ICs did not exceed the acceptable level of 1, thus indicating a negligible non-carcinogenic toxic risk. The values of $HQ_{ing}$ in OS were $6.91 \times 10^0$ owing to the extremely high concentrations of Zn, Pb and Sb, thereby indicating a potential non-carcinogenic health risk (Table 8a). This value was higher than the non-carcinogenic risk value from road dust ingestion reported in other polluted cities. Note that the size of the targeted PM for $HQ_{ing}$ was 20 μm, but the $HQ_{ing}$ values of the finest fraction in road dust in Poland have been reported as $2.97 \times 10^0$ in Warszawa and $2.88 \times 10^0$ in Krakow [84].

**Table 8.** Health risk assessment results. (**a**) Non-carcinogenic risks, (**b**) carcinogenic risk.

**(a)**

| | SH | BS | GS | CW | DB | PH | US | GY | OS | Mean |
|---|---|---|---|---|---|---|---|---|---|---|
| $HQ_{ing}$ | $6.57 \times 10^{-1}$ | $3.87 \times 10^{-1}$ | $1.82 \times 10^{-1}$ | $2.23 \times 10^{-1}$ | $8.51 \times 10^{-2}$ | $3.70 \times 10^{-1}$ | $4.79 \times 10^{-1}$ | $1.46 \times 10^{-1}$ | $6.91 \times 10^{0}$ | $9.07 \times 10^{-1}$ |
| $HQ_{inh}$ | $7.44 \times 10^{-6}$ | $1.93 \times 10^{-5}$ | $8.65 \times 10^{-6}$ | $1.12 \times 10^{-5}$ | $5.39 \times 10^{-6}$ | $2.74 \times 10^{-5}$ | $6.44 \times 10^{-6}$ | $1.45 \times 10^{-5}$ | $2.98 \times 10^{-5}$ | $1.40 \times 10^{-5}$ |
| $HQ_{dermal}$ | $1.65 \times 10^{-4}$ | $2.53 \times 10^{-4}$ | $5.13 \times 10^{-5}$ | $1.26 \times 10^{-4}$ | $3.60 \times 10^{-5}$ | $2.41 \times 10^{-4}$ | $2.05 \times 10^{-4}$ | $5.80 \times 10^{-5}$ | $4.96 \times 10^{-3}$ | $5.60 \times 10^{-4}$ |
| HI | $6.57 \times 10^{-1}$ | $3.87 \times 10^{-1}$ | $1.82 \times 10^{-1}$ | $2.23 \times 10^{-1}$ | $8.52 \times 10^{-2}$ | $3.70 \times 10^{-1}$ | $4.79 \times 10^{-1}$ | $1.46 \times 10^{-1}$ | $6.91 \times 10^{0}$ | $9.08 \times 10^{-1}$ |

**(b)**

| | | SH | BS | GS | CW | DB | PH | US | GY | OS | Mean |
|---|---|---|---|---|---|---|---|---|---|---|---|
| $CR_{ing}$ | As | $1.75 \times 10^{-4}$ | $1.96 \times 10^{-4}$ | $1.07 \times 10^{-4}$ | $9.46 \times 10^{-5}$ | $6.49 \times 10^{-5}$ | $4.77 \times 10^{-4}$ | $5.58 \times 10^{-4}$ | $1.05 \times 10^{-4}$ | $1.02 \times 10^{-2}$ | $1.09 \times 10^{-3}$ |
| | Cd | $4.22 \times 10^{-5}$ | $9.25 \times 10^{-5}$ | $1.61 \times 10^{-5}$ | $4.61 \times 10^{-5}$ | $1.33 \times 10^{-5}$ | $9.23 \times 10^{-5}$ | $7.32 \times 10^{-5}$ | $2.13 \times 10^{-5}$ | $1.89 \times 10^{-3}$ | $2.09 \times 10^{-4}$ |
| | Ni | $8.07 \times 10^{-5}$ | $4.01 \times 10^{-4}$ | $3.76 \times 10^{-5}$ | $8.84 \times 10^{-5}$ | $1.86 \times 10^{-5}$ | $1.00 \times 10^{-4}$ | $3.32 \times 10^{-5}$ | $1.02 \times 10^{-4}$ | $1.02 \times 10^{-4}$ | $1.09 \times 10^{-4}$ |
| | Pb | $5.91 \times 10^{-5}$ | $2.63 \times 10^{-5}$ | $1.14 \times 10^{-5}$ | $1.32 \times 10^{-5}$ | $3.24 \times 10^{-6}$ | $1.59 \times 10^{-5}$ | $2.51 \times 10^{-5}$ | $5.68 \times 10^{-6}$ | $3.45 \times 10^{-4}$ | $5.01 \times 10^{-5}$ |
| $CR_{inh}$ | As | $1.90 \times 10^{-11}$ | $2.12 \times 10^{-11}$ | $1.16 \times 10^{-11}$ | $1.03 \times 10^{-11}$ | $7.04 \times 10^{-12}$ | $5.18 \times 10^{-11}$ | $6.06 \times 10^{-11}$ | $1.14 \times 10^{-11}$ | $1.105 \times 10^{-9}$ | $1.18 \times 10^{-10}$ |
| | Cd | $1.54 \times 10^{-11}$ | $3.36 \times 10^{-12}$ | $5.87 \times 10^{-13}$ | $1.68 \times 10^{-12}$ | $4.83 \times 10^{-13}$ | $3.35 \times 10^{-12}$ | $2.66 \times 10^{-12}$ | $7.75 \times 10^{-13}$ | $6.864 \times 10^{-11}$ | $7.58 \times 10^{-12}$ |
| | Co | $7.82 \times 10^{-11}$ | $4.84 \times 10^{-11}$ | $7.05 \times 10^{-11}$ | $3.29 \times 10^{-11}$ | $1.5 \times 10^{-11}$ | $2.96 \times 10^{-11}$ | $3.08 \times 10^{-11}$ | $2.93 \times 10^{-11}$ | $5.215 \times 10^{-11}$ | $4.32 \times 10^{-11}$ |
| | Ni | $6.52 \times 10^{-12}$ | $3.24 \times 10^{-11}$ | $3.04 \times 10^{-12}$ | $7.14 \times 10^{-12}$ | $1.5 \times 10^{-12}$ | $8.09 \times 10^{-12}$ | $2.68 \times 10^{-12}$ | $8.26 \times 10^{-12}$ | $8.277 \times 10^{-12}$ | $8.82 \times 10^{-12}$ |
| | Pb | $5.12 \times 10^{-12}$ | $2.28 \times 10^{-12}$ | $9.85 \times 10^{-13}$ | $1.15 \times 10^{-12}$ | $2.8 \times 10^{-13}$ | $1.38 \times 10^{-12}$ | $2.17 \times 10^{-12}$ | $4.92 \times 10^{-13}$ | $2.989 \times 10^{-11}$ | $4.34 \times 10^{-12}$ |
| $CR_{dermal}$ | As | $3.99 \times 10^{-6}$ | $4.46 \times 10^{-6}$ | $2.44 \times 10^{-6}$ | $2.16 \times 10^{-6}$ | $1.48 \times 10^{-6}$ | $1.09 \times 10^{-5}$ | $1.27 \times 10^{-5}$ | $2.40 \times 10^{-6}$ | $2.32 \times 10^{-4}$ | $2.49 \times 10^{-5}$ |
| | Pb | $1.35 \times 10^{-6}$ | $6.00 \times 10^{-7}$ | $2.59 \times 10^{-7}$ | $3.02 \times 10^{-7}$ | $7.38 \times 10^{-8}$ | $3.62 \times 10^{-7}$ | $5.73 \times 10^{-7}$ | $1.29 \times 10^{-7}$ | $7.87 \times 10^{-6}$ | $1.14 \times 10^{-6}$ |

🟧: not acceptable risk.

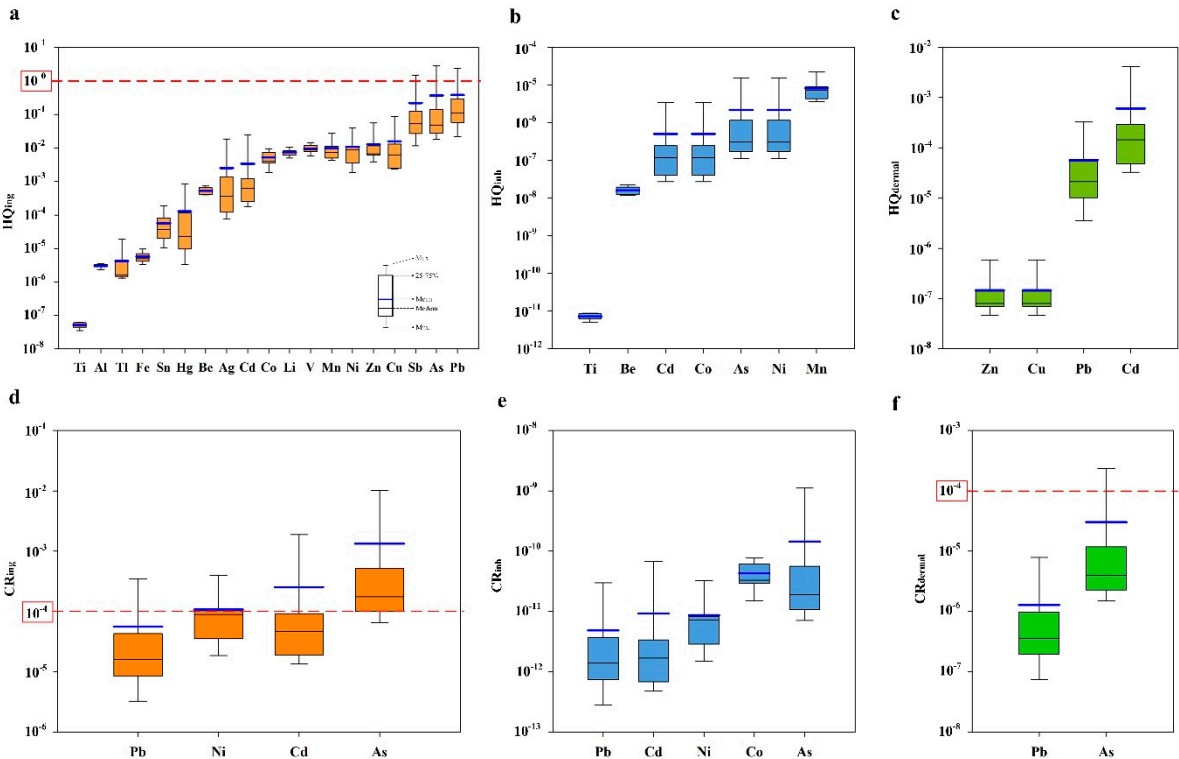

**Figure 4.** Non-carcinogenic risk and carcinogenic risk through ingestion, inhalation, and dermal contact with exposure of HMs from PM10 in RDS, Red line is the acceptable level ($HQ_{ing}$ < 1 ($10^0$), $CR_{ing}$ < $10^{-4}$). (**a**) $HQ_{ing}$, (**b**) $HQ_{inh}$, (**c**) $HQ_{dermal}$, (**d**) $CR_{ing}$, (**e**) $CR_{inh}$, (**f**) $CR_{dermal}$.

The estimated non-carcinogenic human health risks related to inhalation of PTEs in on-road PM10 were Mn > Ni > As > Co > Cd > Be > Ti, indicating that the risk levels of these PTEs were negligible (Figure 4b). The values of $HQ_{inh}$ in the investigated ICs increased in the following order: DB < SH < GS < CW < GY < BS < PH < OS. OS, PH and BS, which ranked highest, had large smelters and steel mills. As mentioned in Section 3.2.1, Mn and Ni, which originated from iron/steelworks and smelters, were deposited on roads and contributed to inhalation toxicity.

The non-carcinogenic health risk ranking of PTEs due to dermal contact with RDS PM10 was Cd > Pb > Cu > Zn (Figure 4c), which showed the same trend as the finest particle fraction of RDS in other regions [84]. The potential non-carcinogenic risk related to dermal contact with RDS PM10 from OS ($HQ_{dermal}$ = 0.00496) was significantly higher than that of other ICs, and the $HQ_{dermal}$ values of on-road PM10 in all other ICs were negligible. The HI is determined by summing the individual HQs. PM10 in road dust was found to have no potential non-carcinogenic effects in any of the ICs other than OS (6.9 × $10^0$).

PTEs were ranked in the order of their estimated carcinogenic human health risk ($CR_{ing}$) related to ingestion as follows: As > Cd > Ni > Pb (Figure 4d). Except for CW and DB, at least one element among As, Cd, Ni and Pb had unacceptable CR levels exceeding 1 × $10^{-4}$. The PM10 of RDS from all nine ICs had no carcinogenic risk for humans related to inhalation ($CR_{inh}$) of PTEs; $CR_{inh}$ values were in the order of As > Co > Ni > Cd > Pb (Figure 4e). The mean carcinogenic risk of As and Pb from dermal contact ($CR_{dermal}$) of PM10 in RDS in this study was less than the acceptable risk level (1 × $10^{-4}$) suggested by the US EPA [83,87,131] (Figure 4f). However, As estimated in OS posed a potential cancer risk to the local population (Table 8b).

These results showed that ingestion of PTEs in on-road PM10 led to the highest risk of carcinogenesis compared to the other pathways.

### 3.3.2. Pollution Sources of Traffic-Origin PTEs and Health Risks

We compared the potential health risks of PTEs originating from traffic activities and those from industrial activities, as identified in the results of Section 3.3. The values of $HQ_{ing}$ of traffic-origin PTEs (As, Ag, Cu, Cd, Hg, Sb, Sn, Pb and Zn) were, on average, nine times (1–102) higher than those of industrial origin (Co, Fe, Mn, Ni, V) (Table 8a). In addition, the HI of the ICs showed a statistical correlation with traffic volume ($p < 0.05$, Supplementary Figure S1a, Table S6). In addition, the Pearson's correlation analysis between the HI of traffic-origin PTEs and traffic volume showed that the non-carcinogenic health risk of ingestion of Pb, Sb and Sn in PM10 was significantly correlated with the traffic volume ($p < 0.05$, Table S7). Regarding carcinogenic risk, the $CR_{ing}$ values of Pb, Cd and As, which are produced from road traffic, were, on average, six times higher than the value of industrial Ni (Table 8b). Among the PTEs, the $CR_{ing}$ of Pb was significantly correlated with traffic volume ($p < 0.01$, Supplementary Figure S1b). Regarding cancer risk through inhalation, the average value of carcinogenic risk of traffic origin PTEs (As, Pb, Cd) was 3.5 times higher than that of the industrial origin PTEs (Ni, Co) (Table 8b).

Pb is one of the major pollutants of RDS in Korea [19,75] and is readily accumulated in biota. Chronic exposure to lead can result in chronic health effects [132], and experimental chronic renal failure in humans [133]. Sb is a toxic element, and $Sb_2O_3$, an important oxide of Sb, is classified as a potential carcinogen in humans [96].

These results suggest that the health risk of road PM10 is related to traffic activity, especially non-exhaust emissions, and traffic volume is also a significant factor in the health risk of on-road PM10.

### 4. Conclusions

Today, road dust accumulated on the road surface is one of the important causative agents of an ambient PM that affects the air quality and human health of cities. This study aimed to investigate the metal composition and pollution degrees of PM10 in road-deposited sediments (on-road PM10) in major industrial complexes (ICs) in Korea and evaluate its human health risks.

This study elucidated that PM10 fractions of road dust from nine ICs were highly contaminated with Sb, Zn, Ag, Pb and Cu, and this contamination was mainly caused by the traffic activities with tire wear and brake dust. The non-carcinogenic and carcinogenic toxicity due to PTE exposure was higher with the elements identified as traffic-origin than those originating from the industry. The carcinogenic risk associated with the ingestion of on-road PM10, calculated Pb, As and Sb exceeded the safe level in ultra-polluted ICs. PTEs that adversely affect human health have been shown to originate mainly from non-exhaust vehicle emissions, and tire and brake pad wear materials are an emerging concern in environmental science.

These results showed that public health risk assessment on PM with less than 10 micrometers on the road should be conducted. In addition, further studies are needed on the risk of PTE and hazardous organic pollutants for which the risk has not yet been identified, such as Sb, and on the source and contribution of pollutants should be considered to create an effective pollution control method.

**Supplementary Materials:** The following are available online at https://www.mdpi.com/article/10.3390/atmos12101307/s1, Figure S1: Correlation between health risk of PM10 and traffic activity (traffic volume), Table S1: Average mass of PM10 per 1 m$^2$ and relative concentration of PM10 fraction in all RDS, Table S2: The results of the same analytical procedure using two types of certified reference materials, MESS-4 and PACS-3 (National Research Council, Ottawa, ON, Canada) Table S3: Specific chemical parameters used in the health risk assessment, Table S4: Coefficients of Kendall's tau b correlation analysis among the $I_{geo}$ of metals in PM10 of RDS, Table S5: Rotated component matrix extracted by principal component analysis (PCA) with Varimax rotation with Kaiser Normalization for heavy metals related to anthropogenic activities in PM10 of RDS, Table S6: Coefficients of Kendall's tau_b correlation analysis among the $I_{geo}$ of metals in PM10 of RDS, Table S7: Pearson's correlation analysis among the $I_{geo}$ of metals in PM10 of RDS.

**Author Contributions:** Conceptualization, J.-Y.C., K.R. and H.J.; methodology, J.-Y.C. and K.R.; Data curation, J.-Y.C., K.R. and H.J.; formal analysis, J.-Y.C., K.R. and H.J.; investigation, K.R. and K.-T.K.; resources, K.R. and K.-T.K.; data curation, J.-Y.C. and K.R.; writing—original draft preparation, J.-Y.C.; writing—review and editing, J.-Y.C., K.R. and H.J.; visualization, J.-Y.C.; project administration, K.R. and K.-T.K.; funding acquisition, K.-T.K. All authors have read and agreed to the published version of the manuscript.

**Funding:** This study was supported by the project "Monitoring of source and behavior of the particulate matter at Busan Seaport area" (PN90430/NRF-2019M1A2A2103955) of the National Research Foundation of Korea (NRF)".

**Institutional Review Board Statement:** Not applicable.

**Informed Consent Statement:** Not applicable.

**Data Availability Statement:** Not applicable.

**Acknowledgments:** We thank Seung-yong Lee for helping us with road dust sampling. And we sincerely thank the three reviewers for their kind and in-depth review of this paper. This paper is funded by the National Research Foundation of Korea (PN90430/NRF-2019M1A2A2103955).

**Conflicts of Interest:** The authors declare no conflict of interest.

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
