# Peer review of "Potentially Toxic Elements (PTEs) Composition and Human Health Risk Assessment of PM10 on the Roadways of Industrial Complexes in South Korea"

_atmosphere, doi:10.3390/atmos12101307_

Round 1

Reviewer 1 Report

General comments

Title: Potentially Toxic Elements (PTEs) composition and Human Health Risk Assessment of PM10 on the roadways of Industrial Complexes in South Korea

The manuscript addresses an interesting issue, by exploring the relation between potentially toxic elements (PTEs) of on-road PM10 in road deposit sediments (RDS) and human health. The sampling was carried out in nine major industrial complexes (ICs) and thus the study provides an adequately descriptive profile of the largest polluted areas of South Korea and the implications in human health of the citizens. The paper is well written and organized while the ratios and statistical tools have been well explained and discussed with the available literature. The source composition derived from the analysis results could help the environmental protection department to formulate correct measures to improve air quality. So, I think this research can be published after revising some points.

Abstract

I would suggest authors to avoid giving so much information about the background of the study and to maintain to the outcomes and results. For example lines 12-16 should be combined to a maximum of 2 sentences. Authors can also rearrange lines 19-24 to the conclusions section and replace them with more informative details about the results and source apportionment.

Introduction

Introduction section is well structured and informative. Some questions-suggestions:

Line 39 Authors may add a more recent reference

Line 44-46 Please define if you refer to electric vehicles or not, according to reference

Line 48-49 Authors should add a reference

Materials –Methods

Line 175 Please add the reference you are referred to (Tomlinson et al. 1980)

Line 177 When you refer to other studies, please add some references.

Line 239 Authors could add some references with other studies using PCA for source apportionment of elements’ composition of PM10 (Hieu et al. 2010, Zhou et al. 2014, Koukoulakis et al. 2019)

Results and Discussion

Table 3.2, 3.3 Authors should revise the values present in tables in order to have the same number of significant digits

Line 309 Please check numbers and references beginning from line 309 as I believe there is a fault in the correct row (e.g. line 309- [75], line 310 – [76], line 311- [75], line 314- [76], line 316- [77], line 318- [78-80] etc, line 323- [81, 82])

Figure 4 Please revise caption to include also CR descriptions. Authors could also resize figure and thus the y axis will be more easily readable

Line 466-467 As I can see in figure 4a authors have put elements in ascending order, not descending.

Line 492-495 In that section, authors have found unacceptable CRing levels for Pb, Ni, Cd, As as shown in Table 7b but in boxplots of figure 4d, values of CRing were observed below the USEPA threshold. Please correct the figure.

Line 510-513 Please check the content and the caption of Tables S5 and S6 to be in accordance with your statements in that lines (E.g. there is no HQing in tables). Also you should change the title of 3.3.2 section to be more suitable for what you present.

Conclusions

Authors conclude their study repeating with many details what they have already presented. I believe that conclusions section should be more precise and not so wide. The whole section should be rewritten in a more concise way so as not to be a summary of the study but a conclusion. Authors should highlight only the most important outcomes of the study without repeating the results section. Most of the conclusions could be a part of the abstract as well.

References

Hieu, N.T., Lee, B.K., 2010. Characteristics of particulate matter and metals in the ambient air from residential area in the largest city in Korea. Atmos. Environ. 98, 526–537.

Zhou, S., Yuan, Q., Li, W., Lu, Y., Zhang, Y., Wang, W., 2014. Trace metals in atmospheric fine particles in one industrial urban city: Spatial variations, sources, and health implications. J. Environ. Sci. 26, 205–213.

Koukoulakis, K.G., Chrysohou, E., Kanellopoulos, P.G., Karavoltsos, S., Katsouras, G., Dassenakis, M., Nikolelis, D., Bakeas, E., 2019. Trace elements bound to airborne PM10 in a heavily industrialized site nearby Athens: seasonal patterns, emission sources, health implications, Atmos. Pollut. Res. 10, 1347–1356, https://doi.org/10.1016/j.apr.2019.03.007

Author Response

Dear reviewer

We thank you very much for your kind review.
Thanks to your advice, we have reviewed the paper more deeply, and it seems that we can improve this manuscript in depth.
The manuscript has been carefully edited, so please review it.

Sincerely yours

Reviewer 2 Report

The paper entitled “Potentially Toxic Elements (PTEs) composition and Human 2

Health Risk Assessment of PM10 on the roadways of Industrial 3 Complexes in South Korea” may be published after the authors will consider below comments.

The paper is very interesting scientifically and it’s great topic. However, there are some missing parts and information such data quality control and assurance, details of the “Analytical Techniques” which the author should add to support the paper. The authors should add most of them to the Supp. Material and leave the important part, figures and tables in the main text.

Please find below specific comments.

Abstract:

Motivation sentence is missing! Clearly write aim of the study!

Please re-write and write brief take-home message for each section. The Authors should write very concisely and deliver your messages/results in short writing and concise paragraph.

  1. Introduction

This section is not interesting in the current format. I would suggest bringing more recent studies in this area and around of world and make a table for them. The author should bring previous study and explain their finding in one table. The table will tell us where is the gap and why this study is important? In addition. Please write the paragraph and compare with previous studies and identify gap research.

Line91: Is this study seriously for the first time??

Finally, the aim of the study should be written in better way. It’s not clear now as the authors mentioned several topics!

  1. Materials and Methods
  • Figure 1 is very general please revise and point out the location etc.
  • 2. Sample Collection and Separation of PM10: This needs to be revised. It’s not interesting in reading, please add more explain and writing comprehensively.
  • 3. Analytical Techniques: Please write more about the techniques and QUALITY CONTROL AND ASSURNACE (This paper cannot be accepted without the QA/QC)
  • Need to mention proper quality control and assurance and how the chemical analysis has been done? Where are the detection limit of elements? How did you use the reference/standard materials? Need proper section in details.
  • How did the authors find Non-carcinogenic and carcinogenic risks assessment for elements? Where are the equations? How they compared with literature review? Authors should adding the details in this section.
  • 6. Statistical Analysis: It needs to add proper equations writing for the ANOVA and PCA calculations.

Please re-write methodology and experimental part, and write down some sub section to make a better structure.

  1. Results and Discussion

This section has some interesting parts, but it’s very long. I would suggest the authors make them shorter and more concise. Some sections should be merged and write a clear story of the topic. All sub-sections are boring to read now and they are need major correction in order to be interesting for readers. Very weak statement without supporting them by previous studies. Please make sure cite the relevant studies and compare with your findings.

  1. Conclusions:

Please re-write and change the structure to bulling points. You can write brief conclusion for each section of the results and discussion. I would suggest write the Limitation/shortcomings of this study and more future work rather than juts “the calculation of PM10 emissions” in this research area.

Author Response

(The authors gave the same response as above.)

Reviewer 3 Report

Dear authors 

This work is good 

Author Response

(The authors gave the same response as above.)

Round 2

Reviewer 2 Report

The Authors have addressed all my comments properly. Well done.